# Laminin N-terminus α31 protein distribution in adult human tissues

**Lee D. Troughton**[1]*, **Raphael Reuten**[2], **Conor J. Sugden**[1], **Kevin J. Hamill**[1]

**1** Institute of Life Course and Medical Sciences, University of Liverpool, Liverpool, United Kingdom,
**2** Biotech Research & Innovation Centre, University of Copenhagen, Copenhagen, Denmark

* leedavid@liverpool.ac.uk

**Data Availability Statement:** All relevant data are within the manuscript and its Supporting information files.

**Funding:** This work was supported by Biotechnology and Biological Sciences Research

## Abstract

Laminin N-terminus α31 (LaNt α31) is a netrin-like protein derived from alternative splicing of the laminin α3 gene. Although LaNt α31 has been demonstrated to influence corneal and skin epithelial cell function, its expression has not been investigated beyond these tissues. In this study, we used immunohistochemistry to characterise the distribution of this protein in a wide-array of human tissue sections in comparison to laminin α3. The data revealed widespread LaNt α31 expression. In epithelial tissue, LaNt α31 was present in the basal layer of the epidermis, throughout the epithelium of the digestive tract, and in much of the epithelium of the reproductive system. LaNt α31 was also found throughout the vasculature of most tissues, with enrichment in reticular-like fibres in the extracellular matrix surrounding large vessels. A similar matrix pattern was observed around the terminal ducts in the breast and around the alveolar epithelium in the lung, where basement membrane staining was also evident. Specific enrichment of LaNt α31 was identified in sub-populations of cells of the kidney, liver, pancreas, and spleen, with variations in intensity between different cell types in the collecting ducts and glomeruli of the kidney. Intriguingly, LaNt α31 immunoreactivity was also evident in neurons of the central nervous system, in the cerebellum, cerebral cortex, and spinal cord. Together these findings suggest that LaNt α31 may be functionally relevant in a wider range of tissue contexts than previously anticipated, and the data provides a valuable basis for investigation into this interesting protein.

## Introduction

Laminins (LMs) are essential extracellular matrix (ECM) structural proteins required for the assembly and function of basement membranes (BMs) [1, 2]. BMs provide cell-ECM interaction sites for epithelial, endothelial, nerve, and muscle cells, as well as providing a substrate for cell migration during tissue remodelling, and signals that define lineage specification, as reviewed in [1–4]. LMs have been studied extensively over the past 40+ years and 12 LM encoding genes have been identified in higher organisms [5]; however, the complexity of the family has further grown with the identification of a series of transcripts encoding non-laminin proteins that are also generated from laminin-encoding genes by alternative splicing. At least four such transcripts have been identified from the 5' end of the LMα3 (LAMA3) and

Council grants BB/L020513/1 and BB/P0257731 (KH), by North West Cancer Research (KH), and Crossley-Barney Bequest to the University of Liverpool (KH), by German Cancer Aid (RR), and the Danish Cancer Society grant R204-A12454 (RR). The funders had no role in study design, data collection and analysis, decision to publish, or preparation of the manuscript.

**Competing interests:** The authors have declared that no competing interests exist.

**Abbreviations:** BM, basement membrane; C-terminus, Carboxyl-terminus; ECM, extracellular matrix; LaNt, Laminin N-terminal; LCC, laminin-coiled coil; LE-repeats, laminin-type epidermal growth factors-like domains; LM, laminin; LN-domain, laminin N-terminal domain; N-terminus, Amino terminus.

LMα5 (LAMA5) genes [6] and at least one of these transcripts is translated into a functional protein, termed Laminin N-terminus α31 (LaNt α31) [6–8].

LaNt α31 is produced from the LAMA3 gene via a process of intron retention and polyadenylation. LAMA3 is unique within the LM family in that is has two promoters giving rise to structurally distinct isoforms; a short LMα3a form and a longer LMα3b. These proteins share common carboxyl-terminal (C-terminal) regions but differ in the length of their amino terminus (N-terminus) [9, 10]. LMα3a expression has been reported as predominantly restricted to squamous epithelia. In contrast, LMα3b, which shares its promoter with LaNt α31, has a much wider distribution profile, although often at lower levels than LMα3a [6, 9–12]. LaNt α31 mRNA (*LAMA3LN1*) expression has been identifed in a wide array of tissues including; the heart, brain, placenta, lung, pancreas, spleen, thymus, prostate, testis, ovaries, small intestine, and in leukocytes as determined by semi-quantitative reverse transcription polymerase chain reaction from whole tissue extracts [6]. At the protein level, LaNt α31 studies have focused on epithelial tissues [6–8]. LaNt α31 protein was identified in the basal layers of the skin and corneal epithelia with localised enrichment in limbal epithelial sub-populations, and in stromal structures including blood vessels. Upregulation of the protein was observed during ex vivo corneal wound repair and in stem cell activation assays, and functional studies using knockdown and overexpression approaches have indicated a role for this comparatively unstudied protein in the regulation of keratinocyte migration and adhesion [6–8].

Although derived from the LAMA3 gene, LaNt α31 is much smaller and does not contain the some of the characteristic LMs features. Specifically, whereas LMs form heterotrimers via a laminin coiled-coil-domain (LCC domain) located toward the C-terminus of each subunit [13–16], LaNt α31 lacks this LCC domain and therefore cannot trimerise. Indeed, as their name suggests, the most striking structural feature of LaNt α31 is a laminin N-terminal domain (LN domain). LN-domains play an essential role in BM assembly by providing the points through which LM to LM interactions occur; a process which involves formation of a ternary nodes between LN-domains between an α, β and γ LM subunits [1, 2, 17–20]. The LaNt α31 LN domain is followed by stretch of laminin-type epidermal growth factor-like domains (LE repeats) and a unique C-terminus which is not conserved with LMs and which does not have any recognised conserved domain architecture [6].

LaNt α31 protein architecture closely resembles other members of the LM superfamily; the netrins, with the notable exception that the netrins have distinct C-terminal domains not present in LaNt α31 [21–24]. Netrins can be broadly classified into two groups; γ-netrins and β-netrins. Netrins-1, 3 and 5 share evolutionary ancestors with the γ laminins [23, 25] and are considered primarily as signalling molecules via binding of their LN domains (domain VI) and LE-repeats (LE2-LE3) to the cell-surface receptors neogenin, deleted in colorectal cancer (DCC), members of the uncoordinated 5 family, and Down-syndrome cell adhesion molecule, as reviewed in [24, 26, 27]. Netrin-4, in contrast, which arose independently from β laminin ancestors [23, 25], appears to be functionally distinct in that it can disrupt LM networks via competition of LM ternary nodes [28], with implications for directed cell migration during angiogenesis and neurogenesis [29–35]. LMs and netrins are known to play context-specific roles across a range of tissues types, with local expression patterns and abundance providing valuable clues to functionality, as reviewed in [2–4, 13, 22, 24, 26, 36–41]. However, the equivalent expression and distribution of LaNt α31 protein have not been determined. The LaNt α31 expression data are of particular interest, as LaNt α31 is likely to play distinct roles in different tissues dependent on local cell-surface receptor and BM make-up.

Here we used LaNt α31 specific monoclonal antibodies to determine the protein distribution across a wide range of human adult tissues for the first time, while comparing to total LMα3 localisation.

## Methods

### Ethical approval

Ethical approval for working with human tissue was conferred by the University of Liverpool Research Ethics Committees (approval number:7488). Microarray sections of 10% neutral buffered formalin-fixed and paraffin-embedded human tissue sections were acquired from Reveal Biosciences (NT02, Reveal Biosciences, San Diego, California, US) and US Biomax (MBN481, US Biomax, Rockville, Maryland, US). Human tongue tissue was acquired from the Liverpool Head and Neck Bioresource (research ethical committee approval number: EC 47.0).

### Antibodies

Mouse monoclonal antibodies (clone 3E11) against residues 437 to 451 of the unique region of LaNt α31 were described previously [7], these were used for immunohistochemistry (IHC) at 0.225 μg mL$^{-1}$, immunoblotting (IB) at 1.8 μg mL$^{-1}$, and enzyme-linked immunosorbent assay (ELISA) at 18 μg mL$^{-1}$. Rabbit polyclonal antibodies raised against a glutathione *S* transferase fusion protein containing the 54 amino acid unique region of LaNt α31 were used as described previously for IHC [6, 7], mouse anti- LMα3a and LMα3b antibodies (mapped to EINSLQSDFT, corresponding to residues within the LCC domain) were used at 50 μg mL$^{-1}$ (clone CL3112, AMAB91123, Sigma-Aldrich, St. Louis, Missouri, US) and mouse IgG for IHC (31903, ThermoFisher Scientific, Waltham, Massachusetts, US).

### Immunohistochemistry

Sections were processed using a Leica Bond autostainer and the Bond™ Polymer Refine Detection system (Leica Biosystems, Wetzlar, Germany). Briefly, sections were dewaxed and rehydrated through a series of xylene and decreasing ethanol concentrations. Antigen retrieval was performed by incubating in Tris/EDTA buffer pH 9 for 20 mins at 60˚C for anti-LaNt α31 antibodies and mouse IgG, or in Tris/citrate buffer pH 6 for 30 mins at 60˚C for anti-LMα3 antibodies. Endogenous peroxidases were blocked with Bond hydrogen peroxide solution for 5 mins at room temperature. Sections were then incubated with primary antibodies at room temperature for 30 mins LaNt α31 and mouse IgG or for 60 mins LMα3) in Bond primary Ab solution (tris-buffered saline [TBS], containing surfactant and protein stabilizer, complete composition not provided). Secondary rabbit anti-mouse IgG Abs with 10% v/v animal serum in TBS were added for 15 mins (LaNt α31 and mouse IgG) or 30 mins (LMα3) at room temperature. Polymer anti-rabbit poly-HRP-IgG Abs (<25 μg mL$^{-1}$) in 10% v/v animal serum in TBS were added at room temperature for 20 mins for LaNt α31 and mouse IgG, or 30 mins for LMα3. 66 mM DAB chromogen substrate was added for 10 mins (LaNt α31 and mouse IgG) or 30 mins (LMα3) at room temperature, then slides were counterstained with 0.1% w/v haematoxylin for 5 mins. At each stage, washes were performed with Bond wash solution (TBS containing surfactant, complete composition not provided), and with deionized water after counterstaining. Sections were dehydrated through a series of ascending ethanol concentrations and xylene, then mounted in Pertex$^{®}$ (Pioneer Research Chemicals Limited, Essex, UK).

Slides were scanned at 20x using an Zeis Axio-slidescanner Z1 equipped with an Axiocam colour CCD camera and processed using ZEN Blue software (all from Carl Zeiss AG, Oberkochen, Germany) and Fiji (ImageJ, U. S. National Institutes of Health, Bethesda, Maryland, US). Figures were prepared using CorelDRAW 2017 (Corel, Ottawa, Canada).

## Short hairpin RNA (shRNA) and LaNt α31 expression construct generation

shRNA sequences targeting the unique portion of the *LAMA3LN1* transcript were designed using the BROAD institute design algorithm (http://www.broadinstitute.org/rnai/public/seq/search). A gblock containing the shRNA sequence (CCCTCTCTCTTCAGAGTATT*)* or non-silencing sequence (TCTCGCTTGGGCGAGAGTAAG), as well as stem loop sequence with *Bam*HI and *Xho*I restriction enzyme compatible overhangs (synthesized by Integrated DNA Technologies, Coralville, Iowa, US), was cloned into the miRNA adapted pGIPz plasmid (Open Biosystems, Huntsvill, Alabama, US). Lentiviral particles were generated as previously described [42] using 293T packaging cells (Gene Hunter Corporation, Nashville, Tennessee, US) with help from the DNA/RNA Delivery Core at Northwestern University (DNA/RNA Delivery Core at Northwestern University, Chicago, Illinois, US).

To generate lentiviral particles for LaNt α31-PAmCherry expression, a gBlock containing the coding sequence for LaNt α31-PAmCherry with *EcoR1* and *NheI* restriction enzyme compatible overhangs (synthesised by Integrated DNA Technologies) was inserted into the pLenti-puromycin vector and packaged in lentiviral particles (produced by Origene). (PS100109, Ori-Gene, Rockville, Maryland, US).

hTCEpi cells [43] cultured in Gibco keratinocyte serum-free medium supplemented 5 ng mL$^{-1}$ EGF, 0.05 mg mL$^{-1}$ BPE, and 0.15 mM CaCl$_2$ (all ThermoFisher Scientific), were seeded at 200,000 cells/well of 6 well plates and transduced with lentiviral particles at final MOI of 0.5 with 8 μg/ml polybrene (Sigma-Aldrich). Transduced cells were selected in 10 μg/ml puromycin (Gibco, ThermoFisher Scientific) for 7 days. Selected cells were seeded at 2.5x 10$^5$ in 6-well plates for 24 hours then scraped into 90 μL urea/sodium dodecyl sulphate (SDS) buffer (10 mM Tris-HCl pH 6.8, 6.7 M Urea, 1% w/v SDS, 10% v/v Glycerol and 7.4 μM bromophenol blue, containing 50 μM phenylmethysulfonyl fluoride (PMSF) and 50 μM N-methylmaleimide (all Sigma-Aldrich), then sonicated and 10% v/v β-mercaptoethanol added (Sigma-Aldrich). Proteins were separated by sodium dodecyl sulfate- polyacrylamide gel electrophoresis using 10% polyacrylamide gels (1.5 M Tris, 0.4% w/v SDS, 10% acrylamide/ bis-acrylamide, [Sigma-Aldrich]), in electrophoresis buffer (25 mM Tris, 190 mM glycine [Sigma-Aldrich], 0.1% w/v SDS, pH 8.5). Proteins were transferred to a nitrocellulose membrane (Bio-Rad Laboratories) using the TurboBlot™ system (Bio-Rad Laboratories) and blocked at room temperature in Odyssey® TBS-Blocking Buffer (Li-COR Biosciences, Lincoln, Nebraska, US) for one hour, then probed overnight at 4˚C with mouse monoclonal antibodies diluted in blocking buffer, washed 3 x 5 mins in phosphate-buffered saline (PBS) with 0.1% Tween-20 (both Sigma-Aldrich) and probed for 1 hour at room temperature in the dark with IRDye® 800CW conjugated goat anti-mouse secondary antibodies (LiCOR Biosciences), diluted in Odyssey® TBS-Blocking Buffer buffer at 0.05 μg mL$^{-1}$. Membranes were then washed for 3 x 5 mins in PBS with 0.1% Tween-20, rinsed with distilled H$_2$O and imaged using the Odyssey® CLX 9120 infrared imaging system (LiCOR Biosciences). Image Studio Light v.5.2 was used to process scanned membranes (https://www.licor.com/bio/products/software/image_studio_lite/).

## Recombinant LaNt α31 production

To guarantee LaNt α31 protein expression and expression tag cleavage, we cloned the maltose-binding protein (MBP) expression enhancer tag [44] followed by a spacer sequence into a modified sleeping beauty vector [45] containing an N-terminal BM40 signal peptide sequence followed by a double Strep II tag. The LaNt α31 gene sequence encoding for aa: 36–488 was generated by polymerase chain reaction and inserted into the multiple cloning site downstream of the MBP sequence. HEK293 cells (ThermoFisher Scientific) were cultured in DMEM/F-12, GlutaMAX™ (ThermoFisher Scientific) containing 10% foetal bovine serum

(Labtech, East Sussex, UK) and seeded at 90% confluency in a 6-well plate overnight, then transfected with the MBP-LaNt construct using FuGENE® HD (Promega, Madiosn, Wisconsin, US). In brief, FuGENE® HD was added to DMEM/F-12, GlutaMAX™ at room temperature for 10 mins, and then incubated with the plasmid DNA mix (transposase vector and LaNt α31 expression vector, ratio 1:9) for additional 20 min. Transfection reagent to plasmid DNA mix ratio was 1:3 (0.5 μg plasmid per well). The transfection DNA mixture was added to the wells and incubated for 24 hrs. Cells were then selected with 4 μg mL$^{-1}$ puromycin (Sigma-Aldrich) for 3 days and expanded prior to seeding into triple flasks (Nunc® Sigma-aldrich). Expression was induced with 1 μg mL$^{-1}$ doxycyclin (Sigma-Aldrich) in 2% serum-containing media when 90% confluency was reached. Media was harvested every 48 hrs for 6 days, filtered, pre-cleared with gelatin sepharose 4B (GE Healthcare, Chicago, Illinois, US) and the MBP-LaNt α31 purified using Strep-Tactin® sepharose 50% suspension (IBA Lifesciences, Göttingen, Germany), eluted with 1.875 mg ml$^{-1}$ desthiobiotin in TBS. The purified protein was dialysed against 1 x PBS and concentration measured on a Nanodrop 2000 (ThermoFisher Scientific) using the molecular weight and the extinction coefficient of the produced protein.

## Enzyme-linked immunosorbent assay

1 μg of MBP-LaNt α31 or a 0.125 to 1 μg dilution series of LM332 or LM511 in PBS (BioLamina AB, Sundbyberg Municipality, Sweden) was added to wells of a 96-well plate (Greiner Bio-One, Kremsmünster, Austria) in duplicate in a final volume of 100 μL and adsorbed overnight at 4˚C. Wells were the blocked 1% BSA in PBS (Sigma-Aldrich) for 2 hours at room temperature, washed, and mouse anti-LaNt α31 antibodies added in PBS for 1 hour at room temperature with rocking, washed, then probed for 1 hr at room temperature with HRP-conjugated goat anti-mouse IgG secondary antibodies 1:500 in PBS (Dako, Agilent Technologies, Santa Clara, California, US). Final washes were performed and the plate developed with 100 μL tetramethylbenzidine substrate for 10 mins (KPL, SeraCare Life Sciences Inc., Milford Massachusetts, US), and stopped with 20 μL 0.5 M sulphuric acid (Sigma-Aldrich). Absorbance was read at 450 nm on a FLUOstar Optima plate reader (B M G Labtech Ltd, Aylesbury, UK). All wash steps were performed 3 x 5 mins with PBS with 0.1% Tween-20.

# Results

## Antibodies validation

The C-terminus of LaNt α31 is unique to this protein and not conserved with any human LMs or netrins (Fig 1A and 1B). Previous publications have validated two separate antibodies that specifically recognise human LaNt α31; rabbit polyclonal antibodies raised against the entire unique region of human LaNt α31 [6], and mouse monoclonal antibodies (clone 3E11) raised against a peptide within the LaNt α31-specific region [7] (Fig 1A and 1B). Consistent with previous findings, near identical distribution of immunoreactivity was observed in human tongue sections processed with these antibodies (Fig 1C). We extended the validation of 3E11 by immunoblotting total protein extracts from immortalised limbal-derived corneal epithelial cells (hTCEPi cells [43]) stably transduced with either shRNAs targeting LaNt α31, scrambled shRNA controls, or full-length LaNt α31-with a PAmCherry tag cDNA. This confirmed that the 3E11 antibodies display affinity for appropriately sized products, whose expression were reduced upon knockdown (Fig 1D). The 3E11 antibodies also recognised recombinant MBP-tagged LaNt α31 but not netrin-4 (Fig 1E). We further confirmed that 3E11 does not display cross-reactivity to LM332 or LM511 in their native forms using ELISA with recombinant LMs and recombinant LaNt α31 protein as positive control (Fig 1E and 1F).

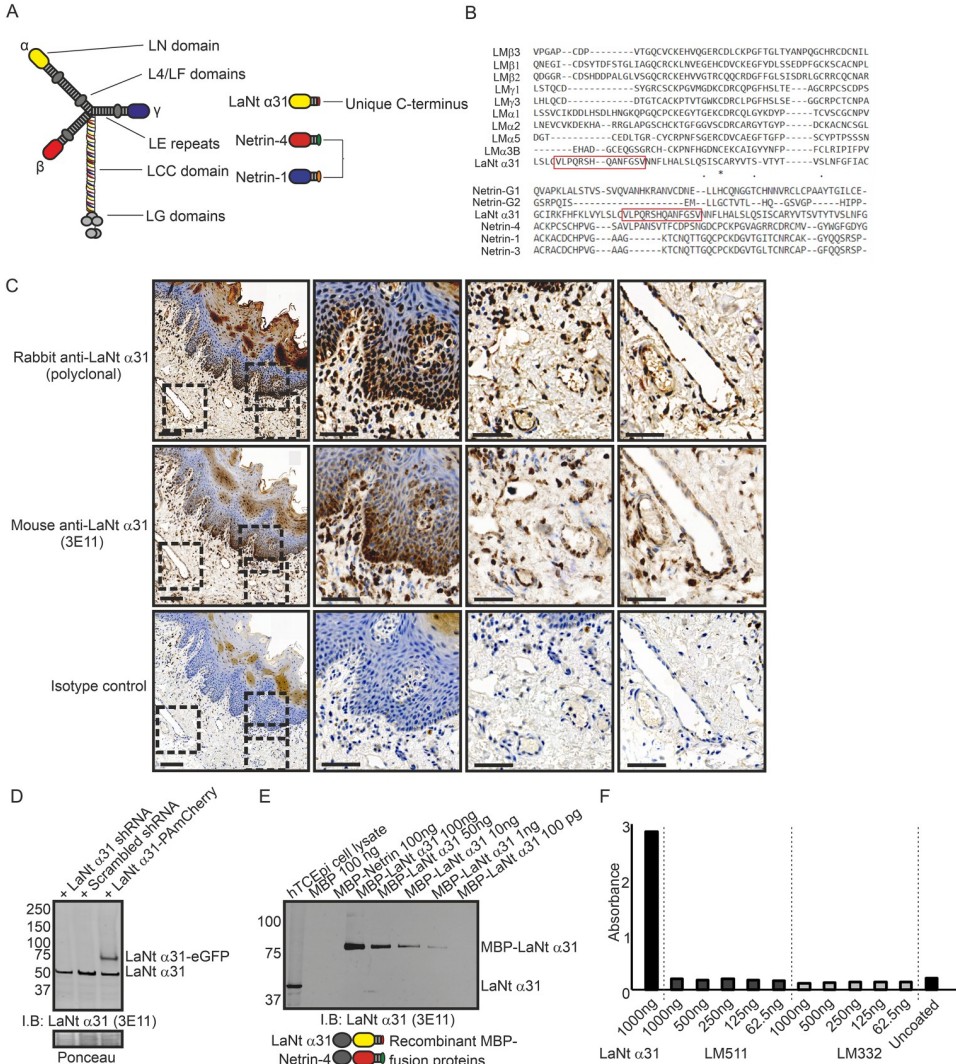

**Fig 1. 3E11 mouse monoclonal antibodies against the unique C-terminus of LaNt α31 validation.** (A) Diagram of the laminin and laminin-related protein architecture. (B) Multiple sequence alignment of LaNt α31 C-terminus with equivalent regions of laminin N-terminal sequences. Peptide unique to LaNt α31 used to generate antibodies highlighted by red box. (C) Paraffin-embedded human tongue tissue sections processed for immunohistochemistry with rabbit polyclonal antibodies against LaNt α31 or mouse monoclonal antibodies against LaNt α31 (3E11). Scale bars represent 200 μm or 100 μm in magnified images. (D) Immunoblot of total protein from hTCEpi expressing either: shRNAs specific to LaNt α31, scrambled shRNA controls, or full-length LaNt α31-PAmCherry cDNA. Ponceau S stained membrane below. (E) Immunoblot of dilution series of recombinant MBP-LaNt α31, and MBP-netrin-4 probed with 3E11 antibodies. Diagram below shows the structure of the recombinant MBP-LaNt α31 and Netrin-4. (F) ELISA using recombinant MBP-LaNt α31, LM511, or LM332. Wells were coated overnight, blocked with BSA, and then probed with 3E11 antibodies. Bound 3E11 antibodies were detected with HRP-conjugated secondary antibodies, HRP-substrate added and absorbance read at 450 nm.

The LaNt α31 protein has been shown to be present in the BM of skin and enriched in basal epithelial cells of the corneal limbus [7], while in situ hybridisation demonstrated that the mRNA (*LAMA3LN1*) is enriched in the basal layer of the epidermis and hair follicles [6]. Using the 3E11 monoclonal antibodies, we observed a similar expression profile. Specifically, LaNt α31 expression was predominantly restricted to the basal layer of the epidermis, with localised enrichment within a sub-population of cells (Fig 2, top row, arrowheads). As

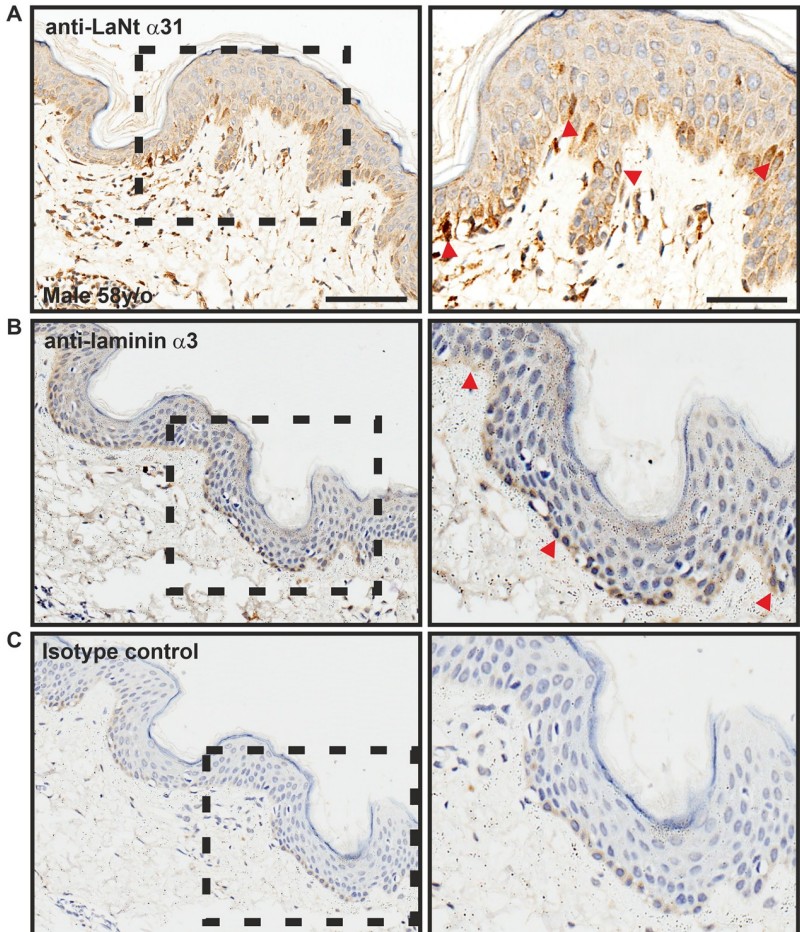

**Fig 2. LaNt α31 is enriched in a sub-population of basal epidermal keratinocytes.** Paraffin-embedded human tissue microarray sections processed for immunohistochemistry with mouse monoclonal antibodies against LaNt α31 (3E11, A), laminin α3 (B), or mouse IgG (C). Scale bars represent 100 μm or 50 μm in magnified images. Arrowheads in (A) indicate localised enrichment of LaNt α31. Arrowheads in (B) indicate basement membrane reactivity for laminin α3.

expected, monoclonal antibodies raised against the LCC domain that is shared between both LMα3a and LMα3b displayed immunoreactivity along BM at the dermal-epidermal junction (Fig 2, middle row, arrowheads). In addition to the epithelial reactivity, both the anti-LaNt α31 and anti-LMα3 antibodies bound to sub-populations of stromal cells (Fig 2).

## LaNt α31 is expressed in the blood vasculature

LaNt α31 displayed immunoreactivity in and around blood vessels in all tissues tested (Fig 3A–3D). Specifically, LaNt α31 immunoreactivity was evident in larger vessels, arterioles and venules, and in the larger capillaries likely in vascular endothelial cells and pericytes. In many cases, particularly for arterioles, the strongest signal was observed in individual cells around the vessel and in vessel walls (Fig 3, arrowheads). The pattern was not uniform to all vasculature; around large venules a distinct ECM / BM-like staining was obtained in the stroma surrounding vessels, with a reticular fibre-like ECM pattern (Fig 3, chevrons). There was no apparent LaNt α31 expression in the lymphatic vessels of these sections (Fig 3, asterisks).

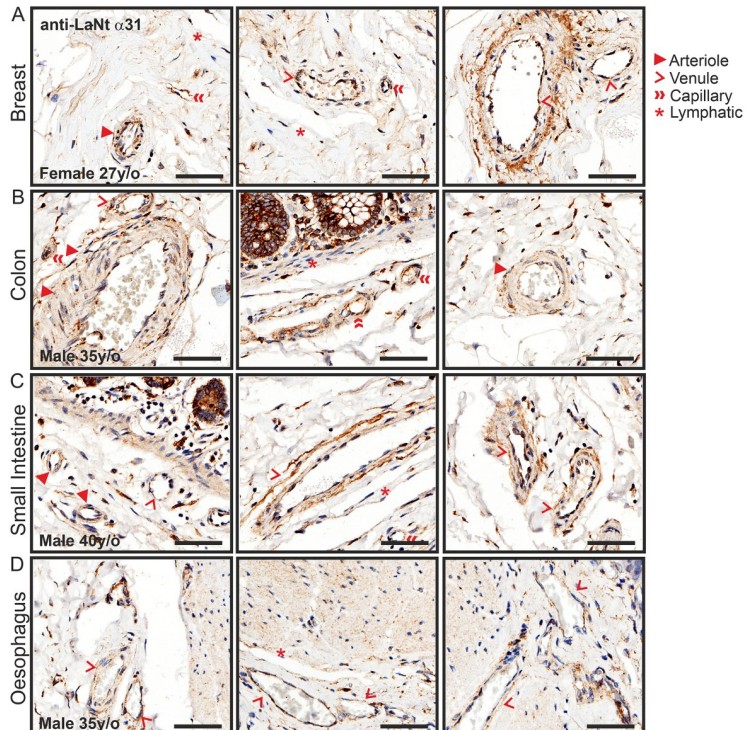

**Fig 3. LaNt α31 is expressed throughout the blood vasculature.** Paraffin-embedded human tissue microarray sections processed for immunohistochemistry with mouse monoclonal antibodies against LaNt α31 (3E11). Representative images taken from: (A) breast, (B) colon, (C) small intestine, (D) oesophagus. Scale bars represent 60 μm.

## LaNt α31 in the breast and lung is assembled into basement membrane structures

In breast tissue, both the inner luminal epithelial and outer myoepithelial cells displayed weak affinity for LaNt α31 antibodies (Fig 4A). Remarkably, the ECM surrounding large collecting ducts of the terminal duct lobular units displayed particularly intense immunoreactivity (Fig 4A, arrowheads), although the smaller lobular acini were negative. These distributions are similar to the reticular fibre-like distribution surrounding larger vessels and ducts (compare Figs 3A and 4A), and resembles that reported for fibulin-2 [46–48]. Although sections from precisely matched tissue were not available for LMα3, in the closest comparison possible, a similar ECM pattern of immunoreactivity was also observed (Fig 4B, arrowheads); however, in contrast to LaNt α31, immunoreactivity for LMα3 was present around the smaller lobular acini (Fig 4B, chevrons). In the lung tissue, LaNt α31 immunoreactivity was observed in the BM underlying the alveolar epithelium that encompass the air sacks (Fig 5A arrowheads), and in similar reticular fibre-like structures (Fig 5A chevrons). In contrast, LMα3 displayed weak immunoreactivity which was restricted to the alveolar BMs (Fig 5B arrowheads).

## LaNt α31 is expressed throughout the epithelium of the digestive tract

LaNt α31 immunoreactivity was detected throughout the oesophagus (Fig 6A), stomach (Fig 6B), small intestine (Fig 6C), colon (Fig 6D), and rectum (Fig 6E), where it displayed a diffuse distribution throughout all layers of the epithelium. LMα3 largely matched the LaNt α31 in all except oesophageal and rectal tissue (Fig 6A and 6E, respectively), where LMα3 was restricted

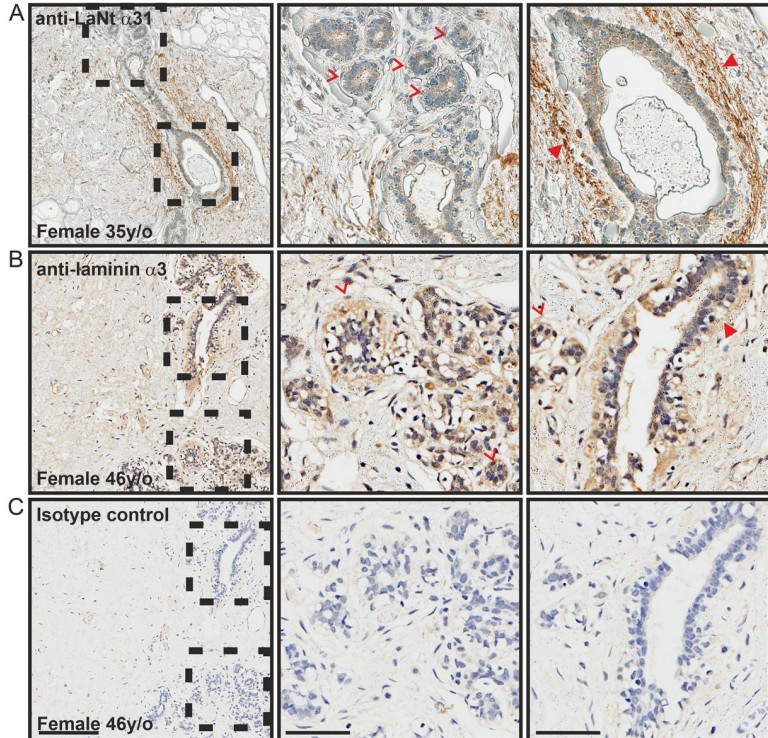

**Fig 4. LaNt α31 is enriched in the ECM surround terminal ducts in breast tissue.** Paraffin-embedded human tissue microarray sections processed for immunohistochemistry with mouse monoclonal antibodies against LaNt α31 (3E11, A), laminin α3 (B), or mouse IgG (C). Arrowheads denote matrix immunoreactivity surrounding ducts, chevrons indicate lobular acini. Scale bars represent 200 μm or 100 μm in magnified images.

to the epithelial BM of the oesophagus (Fig 6A, arrowhead), or extremely weak in the rectal epithelium. In addition to the vasculature, the LaNt α31 and LMα3 antibodies both also labelled a sub-population of stromal cells in all tissue throughout the digestive tract.

## LaNt α31 is enriched in specific cell sub-populations in kidney, pancreas, liver and spleen

In the kidney, LaNt α31 immunoreactivity was observed in the cuboidal epithelium of proximal and distal tubules and collecting ducts, interestingly the intensity was markedly different between structures; with some ducts showing particularly high reactivity (Fig 7A, arrowheads). LaNt α31 immunoreactivity was also observed throughout the parietal epithelium of the Bowman's capsule (Fig 7A, chevron), and in a sub-population of cells in the glomeruli and glomeruli-surrounding stroma. LMα3 largely matched LaNt α31 in the tubules, collecting ducts, Bowman's capsule and glomeruli, but only displayed weak immunoreactivity in some stromal cells, and its intensity did not vary as markedly between ductal structures (Fig 7A).

LaNt α31 immunoreactivity in the liver was observed throughout the cords of hepatocytes (Fig 7B), with a higher intensity in the branches of the bile ducts (Fig 7B, arrowheads). No LaNt α31 was observed in the vascular sinusoid endothelium between hepatocyte cords, nor in stromal tissue, including the accompanying Kupffer cells and fibroblasts. Sections from precisely matched tissue location were not available for LMα3; however, in the closest comparative section, a similar distribution of weak immunoreactivity for LMα3 was observed in the hepatocyte cords and negative the stromal cells (Fig 7B).

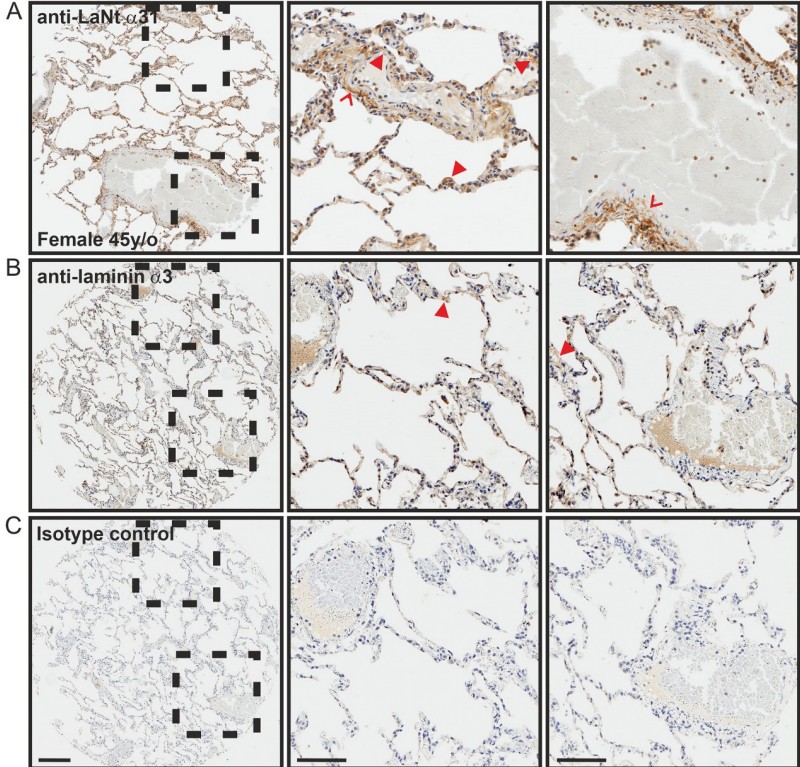

**Fig 5. LaNt α31 localises to the alveolar basement membranes in lung tissue.** Paraffin-embedded human tissue microarray sections processed for immunohistochemistry with mouse monoclonal antibodies against LaNt α31 (3E11, A), laminin α3 (B), or mouse IgG (C). Arrowheads depicts basement membrane, chevrons indicate reticular fibre-like immunoreactivity. Scale bars represent 200 μm or 100 μm in magnified images.

In the pancreas, LaNt α31 was found in the acini epithelium, enriched around the basal layer towards the outer edge (Fig 7C). LMα3 immunoreactivity was also present in the same locations, with basal enrichment (Fig 7C).

In the spleen, LaNt α31 was found throughout the tissue. In the red pulp, the most notable immunoreactivity was observed in the splenic cords of Billroth (Fig 7D, arrowheads), while venous sinuses were comparatively weak (Fig 7D). In the white pulp, comprised of lymphatic tissue, reactivity was much more intense in a sub-population of immune cells, while negative in others. This distribution pattern was largely paralleled by LMα3 (Fig 7D).

## LaNt α31 is widely expressed in reproductive organs

LaNt α31 immunoreactivity was found throughout the female and male reproductive systems. In the ovaries, patchy distribution was observed in the follicular cells of the primordial follicles (Fig 8A, arrowheads). Weak reactivity was also observed in the primary follicles (Fig 8A, chevrons), and intense reactivity in the vasculature (Fig 8A, double chevrons). LMα3 immunoreactivity was also observed in the same structures (Fig 8A: primary follicles, chevrons; vasculature, double chevrons), but was more uniform in distribution in the follicular cells of the primordial follicles (Fig 8A, arrowheads). In the fallopian tube tissue, both LaNtα31 (Fig 8B) and LMα3 (Fig 8B) were found throughout uterine epithelium; however, the intensity of the LaNt α31 reactivity was notably higher in peg cells compared to ciliated cells (Fig 8B arrowheads and chevrons, respectively). Very similar distribution for both LaNt α31 and LMα3

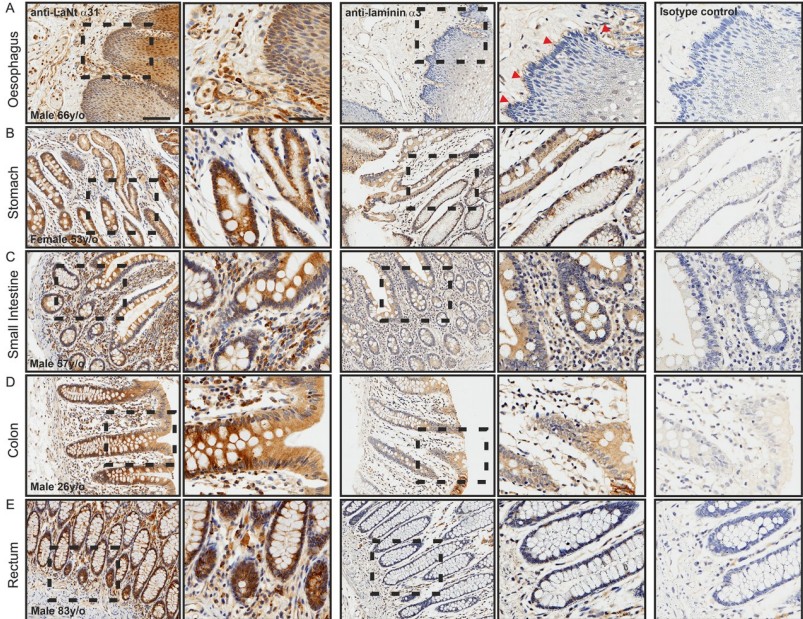

**Fig 6. LaNt α31 is expressed throughout the epithelium of the digestive tract.** Paraffin-embedded human tissue microarray sections processed for immunohistochemistry with mouse monoclonal antibodies against LaNt α31 (3E11), laminin α3, or mouse IgG. (A) oesophagus, (B) stomach, (C) small intestine, colon (D), rectum (E). Arrowheads in (A) indicate basement membrane immunoreactivity. Scale bars represent 100 μm or 50 μm in magnified images.

immunoreactivity was observed in the uterine gland of the endometrium (Fig 8C, arrowheads). LaNt α31 in the cervix was relatively weak compared to the other reproductive tissues. However, LaNt α31 again localised to reticular fibre-like structures in the stromal tissue (Fig 8D, arrowhead), with a similar distribution as in lung tissue (compare with Fig 5A). This

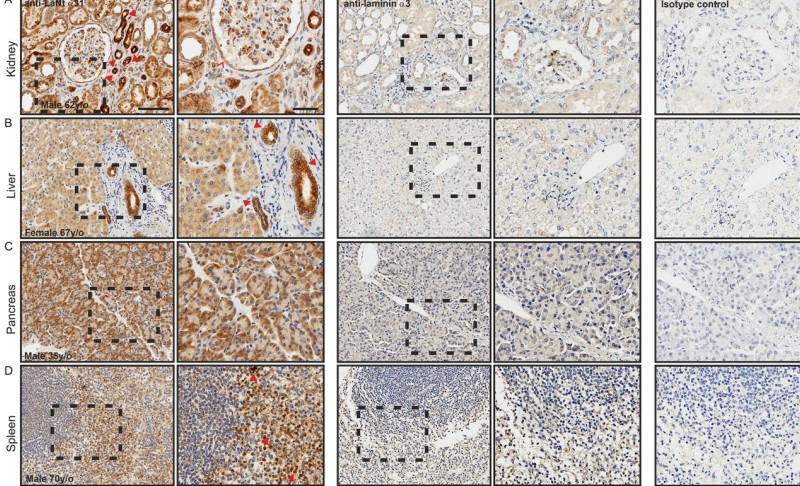

**Fig 7. LaNt α31 expression in kidney, pancreas, liver, and spleen, is largely similar to that of LMα3.** Paraffin-embedded human tissue microarray sections processed for immunohistochemistry with mouse monoclonal antibodies against LaNt α31 (3E11), laminin α3, or mouse IgG. (A) kidney, (B) liver, (C) pancreas, (D) spleen. Arrowheads in (A) depict ducts displaying high immunoreactivity, chevrons indicate parietal epithelium of the Bowman's capsule. Arrowheads in (B) indicate branches of the bile ducts. Arrowheads in (D) indicate splenic cords of Billroth. Scale bars represent 100 μm or 50 μm in magnified images.

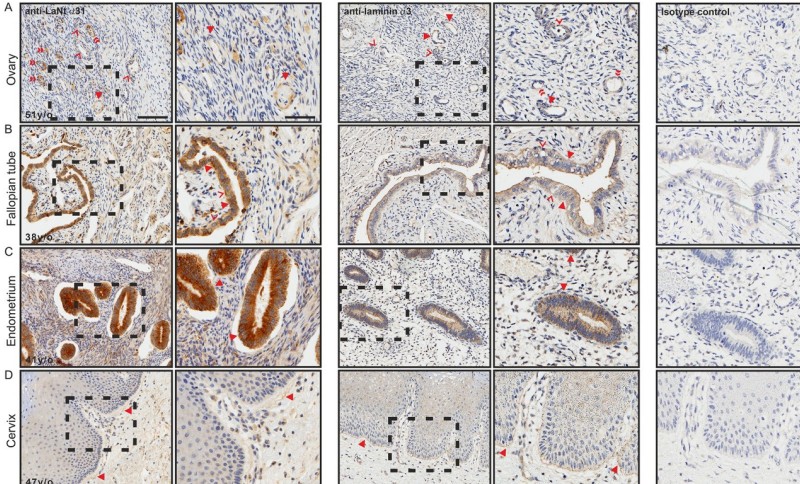

**Fig 8. LaNt α31 is expressed throughout the female reproductive system.** Paraffin-embedded human tissue microarray sections processed for immunohistochemistry with mouse monoclonal antibodies against LaNt α31 (3E11), laminin α3, or mouse IgG. (A) ovary, (B) fallopian tube, (C) endometrium, (D) cervix. Arrowheads in (A) indicate follicular cells of the primordial follicles, chevrons indicate primary follicles, and double chevrons indicate vasculature. Arrowheads in (B) indicate peg cells, chevrons indicate ciliated cells. Arrowheads in (C) indicates uterine glands of the endometrium. Arrowheads in (D) indicate reticular fibre-like immunoreactivity. Scale bars represent 100 μm or 50 μm in magnified images.

reticular fibre-like distribution of LaNt α31 was in contrast to the stratified squamous epithelium BM structure recognised by the LMα3 antibodies (Fig 8D).

In the testis, LaNt α31 was present throughout the seminiferous epithelium. Weak reactivity was observed in the Sertoli cells (Fig 9A, chevrons), with more intense signal from a sub-population of the spermatogonia (Fig 9A, arrowheads). A BM-like distribution was observed under the smooth muscles cells that encompass the seminiferous epithelium (Fig 9A, double chevrons), while the smooth muscle layers themselves were negative. Although sections from precisely matched tissue were not available for LMα3, similar BM-like immunoreactivity was evident, with some weak reactivity present within the seminiferous epithelium in the closest comparison possible, (Fig 9A).

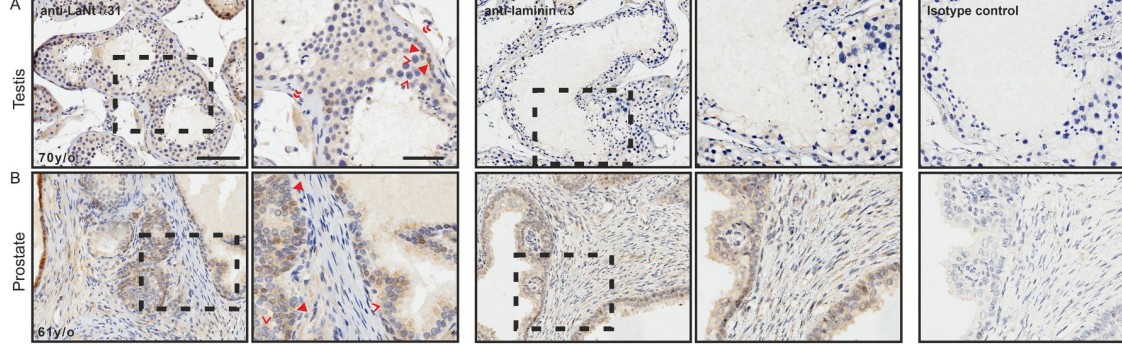

**Fig 9. LaNt α31 is expressed in the testis and prostrate.** Paraffin-embedded human tissue microarray sections processed for immunohistochemistry with mouse monoclonal antibodies against LaNt α31 (3E11), laminin α3, or mouse IgG. (A) testis, (B) prostrate. Arrowheads in (A) indicate a sub-population of the spermatogonia, chevrons indicate Sertoli cells. Arrowheads in (B) indicate basal layer of the acini, chevron indicate acinar cells. Scale bars represent 100 μm or 50 μm in magnified images.

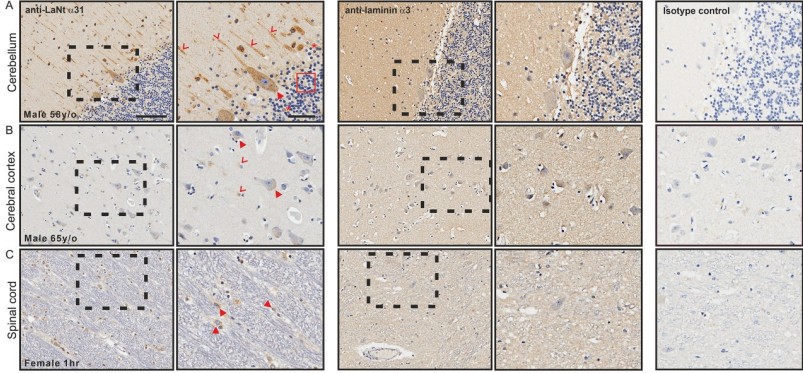

**Fig 10. LaNt α31 is expressed by neurons in the central nervous system.** Paraffin-embedded human tissue microarray sections processed for immunohistochemistry with mouse monoclonal antibodies against LaNt α31 (3E11), laminin α3, or mouse IgG. (A) cerebellum, (B) cerebral cortex, (C) spinal cord. Arrowheads in (A) indicate Purkinje neurons in the ganglionic layer, chevrons indicate axons throughout granular layer. Arrowheads in (B) indicate cell bodies towards the axon hillock of pyramidal cells, chevrons indicate glial cells. Arrowheads in (C) indicate motor neurons. Scale bars represent 100 μm or 50 μm in magnified images.

In the prostate, LaNt α31 immunoreactivity was observed throughout the acini (Fig 9B), however, intensity was stronger in the basal cells (Fig 9B, arrowheads) compared to the acinar cells (Fig 9B, chevrons). LMα3 was also observed throughout the acini (Fig 9B), although the intensity was more uniform.

## LaNt α31 is expressed in neurons of the central nervous system

In the cerebellum, LaNt α31 immunoreactivity was observed in Purkinje neurons in the ganglionic layer (Fig 10A, arrowheads) and axons throughout granular layer (Fig 10A, chevrons), whereas the granule cells in the granular layer, with the exception of a few cells, were negative (Fig 10A, red box). In the cerebral cortex, LaNt α31 immunoreactivity was observed in the cell bodies towards the axon hillock of pyramidal cells (Fig 10B, arrowheads) and in some glial cells (Fig 10B, chevrons). This distribution was striking similar to that observed in the spinal cord tissue taken from a new-born, were LaNt α31 was evident in the motor neurons, but appeared throughout the entire cell body and not just toward the axon hillock (Fig 10C, arrowheads). LMα3 was not detected in either the cerebellum (Fig 10A), cerebral cortex (Fig 10B), nor spinal cord (Fig 10C).

## Discussion

The findings reported here demonstrate that LaNt α31 is widely distributed across multiple tissues including the epithelia, vascular and stromal cells throughout most tissues, and in neurons in the central nervous system. Moreover, that LaNt α31 is present in distinct structures and generally more widespread than LMα3 across the tissues tested, despite their shared genetic origins. Surprisingly, in most situations the strongest LaNt α31 signal was obtained from cells rather than BM structures; however, ECM enrichment in reticular fibre-like structures were frequently observed, particularly surrounding large vessels, terminal ducts in the breast, and alveolar sacks in the lungs. These findings naturally lead to questions regarding the role of LaNt α31 in these different structures and contexts.

As LaNt α31 and LMα3b share a promoter, we predicted a broadly similar distribution pattern between the two proteins. Consistent with this, LMα3b has been described in the BM at the dermal-epidermal junction of skin and underlying the epithelium of the oesophagus,

breast, lungs, and the endothelium of vessels, and the LaNt α31 immunoreactivity was also strong in these regions [12, 49, 50]. For the other tissues examined, there are no published data available at the protein level. However, at the transcript level, *LAMA3LN1* and *LAMA3B* have both been described as having widespread expression, although were not detected in liver, kidney, or spleen tissue-derived cDNAs [6, 11]. *LAMA3B* was reported as being highly enriched in the developing mouse brain [51] but expressed at extremely low levels in adult human brain [6]. The latter agree with the findings here of apparent absence of LMα3 but positive LaNt α31 immunoreactivity in the neurons throughout the CNS. Together, these collective data suggest differences between the expression, stability, and turnover or the mRNA and protein, and imply that changes in LAMA3 splicing behaviour occurs during development and ageing.

The distinct tissue expression and localisation profiles identified here for LaNt α31 suggests that it could play context-specific roles. When considering the LaNt α31 findings, the LMs present in each BM may be of particularly relevance as LaNt α31 is essentially a biologically-active α-type LM LN-domain, which could potentially bind to β and γ LN-domain complexes [49]. LaNt α31 may behave differently in tissues with BMs that contain predominantly a single LN-domain LMs, such as the epidermis and squamous epithelium of the digestive tract which contain predominantly LM3a32, compared with two LN-domain LMs, such as the endothelium in small vessels where LM411 is abundant, and compared again to those tissues where three LN-domain LMs predominate, such as the kidney glomeruli, as reviewed in [3, 4, 52–54]. In this interpretation, clues as to LaNt α31 function are suggested by the structural similarities with netrin-4, which binds with high affinity to the LMγ1 subunit, and can disrupt LM networks [28, 55]. However, the affinity of the LaNt α31 LN-domain for LN domains is much lower than netrin-4 [28, 49], and therefore LaNt α31 could only achieve the same disruptive functionality if present at very high concentrations. However, one could envisage situations where local partial network disruption could be beneficial, for example, in amoeboid-style migration through BMs such as immune cell infiltration through the blood vessel BMs [56–58].

Context-specific signalling roles for LaNt α31 could also be possible. LN-domain containing LM fragments have repeatedly been implicated in integrin-mediated signalling, with the proteolytically released LN-domain containing fragment from LMα3b being shown to support keratinocyte adhesion and proliferation via α3β1 or α6β1 integrins [59], while the α1 LN-domain containing short-arm of LM111 can bind integrins α1β1 and α2β1, albeit at low affinity [60]. Signalling from a LN-domain containing fragment from LMβ1 has been shown to induce a shift from epithelial to mesenchymal-related gene expression via α3β1 integrin [61]. While integrin-mediated signalling is most likely, netrin-like signalling via classical netrin receptors cannot be ruled out. Netrins are well established as being central mediators of numerous biological processes including; neurogenesis, lymphangiogenesis, and haemangiogenesis [29–34, 62]. The expression of LaNt α31 in neurons, throughout the blood but not lymphatic vasculature puts it in appropriate locations to influence similar processes either directly as a signalling molecule in its own right, or indirectly by modulating netrin-mediated signalling.

## Conclusions

LaNt α31 is an interesting protein that is much more widely expressed than previously thought, with the differences in localisation and distribution suggesting a context specificity of function. These findings provide a platform for future functional studies were ECM remodelling is a hallmark, such as during development and ageing, and in pathological situations such as chronic wound repair and cancer.

## Supporting information

**S1 Raw Images.**
(TIF)

## Acknowledgments

The authors would like to thank the donors, without whom this work would not be possible.

## Author Contributions

**Conceptualization:** Lee D. Troughton, Kevin J. Hamill.

**Data curation:** Lee D. Troughton.

**Formal analysis:** Lee D. Troughton, Conor J. Sugden, Kevin J. Hamill.

**Funding acquisition:** Kevin J. Hamill.

**Investigation:** Lee D. Troughton.

**Methodology:** Lee D. Troughton, Raphael Reuten.

**Project administration:** Lee D. Troughton.

**Resources:** Raphael Reuten.

**Supervision:** Kevin J. Hamill.

**Writing – original draft:** Lee D. Troughton, Kevin J. Hamill.

**Writing – review & editing:** Lee D. Troughton, Raphael Reuten, Conor J. Sugden, Kevin J. Hamill.

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
