## [Decision Letter · Decision Letter 0]

16 Sep 2020

Laminin N-terminus α31 protein distribution in adult human tissues

PONE-D-20-17430

Dear Dr. Troughton,

We’re pleased to inform you that your manuscript has been judged scientifically suitable for publication and will be formally accepted for publication once it meets all outstanding technical requirements.

Kind regards,

Jo-Ann L. Stanton

Academic Editor

PLOS ONE

Additional Editor Comments (optional):

This is nicely presented work describing the distribution of an alternative splice variant of the laminin 3alpha gene. I would recommend the authors review the manuscript once more to correct the typographical errors. I would also like to direct you to the comments made by Reviewer 2 regarding laminin 3alpha gene knockout phenotype. Though not required for this manuscript it is worth considering for future work.

Reviewers' comments:

Reviewer's Responses to Questions

**Comments to the Author**

1. Is the manuscript technically sound, and do the data support the conclusions?

Reviewer #1: Yes

Reviewer #2: Yes

2. Has the statistical analysis been performed appropriately and rigorously? 

Reviewer #1: N/A

Reviewer #2: N/A

3. Have the authors made all data underlying the findings in their manuscript fully available?

Reviewer #1: Yes

Reviewer #2: Yes

4. Is the manuscript presented in an intelligible fashion and written in standard English?

Reviewer #1: Yes

Reviewer #2: Yes

5. Review Comments to the Author

Reviewer #1: Several years ago Dr. Hamill described LnNT a31, a netrin-like protein splice variant of the laminin alpha3 subunit. In the current submission, the authors describe the tissue distribution of the protein with specific antibodies. They report that the protein is widely distributed in the epithelia and vasculature of different tissues, and also is found in CNS neurons. The antibody-stained images presented are of high quality. While the protein under study shares the same promoter as the a3 subunit found in laminins within basement membranes, it is interesting to note that LnNT a31 is particularly enriched within the secretory cells and thus there is extension beyond the confines of the basement membranes. The study raises questions about, and helps set the stage for determining the specific functions of this novel protein.

Reviewer #2: The authors of a manuscript entitled: “Laminin N-terminus α31 protein distribution in adult human tissues” present their results on tissue distribution of a unique fragment of laminin 332. The presented study is well designed and carefully executed. The authors provide strong evidence for the specificity of antibodies they used in the study. Although the research is exclusively descriptive, it offers valuable information about a specific split variant of the LAMA3 gene.

It would be of value, however, if the authors discussed their findings in the context of published literature describing the effects of the knock-out of the Lama3 gene. It is expected that knocking out this gene would also have an impact on tissues and organs in which the authors detected LaNt α31.

6. PLOS authors have the option to publish the peer review history of their article (what does this mean?). If published, this will include your full peer review and any attached files.

Reviewer #1: **Yes: **Peter D. Yurchenco

Reviewer #2: No

---

## [Editor Report · Acceptance letter]

20 Nov 2020

PONE-D-20-17430 

Laminin N-terminus a31 protein distribution in adult human tissues 

Dear Dr. Troughton:

I'm pleased to inform you that your manuscript has been deemed suitable for publication in PLOS ONE. Congratulations! Your manuscript is now with our production department. 

Kind regards, 

on behalf of

Dr. Jo-Ann L. Stanton 

Academic Editor

PLOS ONE